# Research on Risky Driving Behavior of Novice Drivers

**Longhai Yang [1], Xiqiao Zhang [1], Xiaoyan Zhu [1], Yule Luo [1,\*] and Yi Luo [2]**

1   School of Transportation Science and Engineering, Harbin Institute of Technology, Harbin 150090, China; yanglonghai@hit.edu.cn (L.Y.); zxq103@126.com (X.Z.); zhuxiaoyan1994@163.com (X.Z.)
2   Traffic Management Research Institute of the Ministry of Public Security, Wuxi 214151, China; luoy_tmri@163.com
\*   Correspondence: 19S032042@stu.hit.edu.cn

**Abstract:** Novice drivers have become the main group responsible for traffic accidents because of their lack of experience and relatively weak driving skills. Therefore, it is of great value and significance to study the related problems of the risky driving behavior of novice drivers. In this paper, we analyzed and quantified key factors leading to risky driving behavior of novice drivers on the basis of the planned behavior theory and the protection motivation theory. We integrated the theory of planned behavior (TPB) and the theory of planned behavior (PMT) to extensively discuss the formation mechanism of the dangerous driving behavior of novice drivers. The theoretical analysis showed that novice drivers engage in three main risky behaviors: easily changing their attitudes, overestimating their driving skills, and underestimating illegal driving. On the basis of the aforementioned results, we then proposed some specific suggestions such as traffic safety education and training, social supervision, and law construction for novice drivers to reduce their risky behavior.

**Keywords:** novice driver; risky driving behavior; theory of planned behavior; protection motivation theory

## 1. Introduction

With the rapid development of China's economy, the number of vehicles in China is dramatically increasing. According to the statistics of the Traffic Management Bureau of the Ministry of Public Security of China, the number of motor vehicles in China was as high as 310 million by the end of 2017, whereas the number of motorists increased to 385 million. At the same time, traffic accidents, especially serious traffic accidents, are still occurring, and traffic safety has become a key issue.

Bad driving behavior is one of the main factors leading to road traffic accidents. According to statistics, if the number of traffic accidents is divided by a driver's number of years of driving experience, the number of traffic accidents for drivers with 1–5 years of driving experience is the largest, accounting for 45.3% of the total. The probability of road traffic accidents caused by novice drivers who lack practical experience and cannot make accurate judgments and responses facing changing traffic conditions is much higher than that of experienced drivers. Novice drivers usually have higher levels of risky and/or bad driving behavior when compared with experienced drivers. Therefore, it is particularly important to explore the bad behavior of novice drivers to improve road traffic safety.

In this paper, we integrate the planned behavior theory (TPB) and the protection motivation theory (PMT) to analyze the bad driving behavior of novice drivers, and study the relationship between the bad driving behavior of novice drivers and their subjective factors, social cognitive factors, and traffic education factors. The results can help explain the risky driving behavior of novice drivers, as well as provide the theoretical basis for correcting bad driving behavior with strategies such as early education and intervention, as well as bonus–punishment differentiation specifically for such individuals. It can

also improve the road safety awareness of novice drivers and their awareness of compliance with traffic rules.

The remainder of the paper is organized as follows: Section 2 presents a literature review on bad driving behavior and traffic accidents; Section 3 introduces the methodology, including the TPB, the PMT, and analysis of factors affecting bad driving behavior; Section 4 gives a case study; and, finally, Section 5 concludes the paper. Figure 1 shows the overview of this paper.

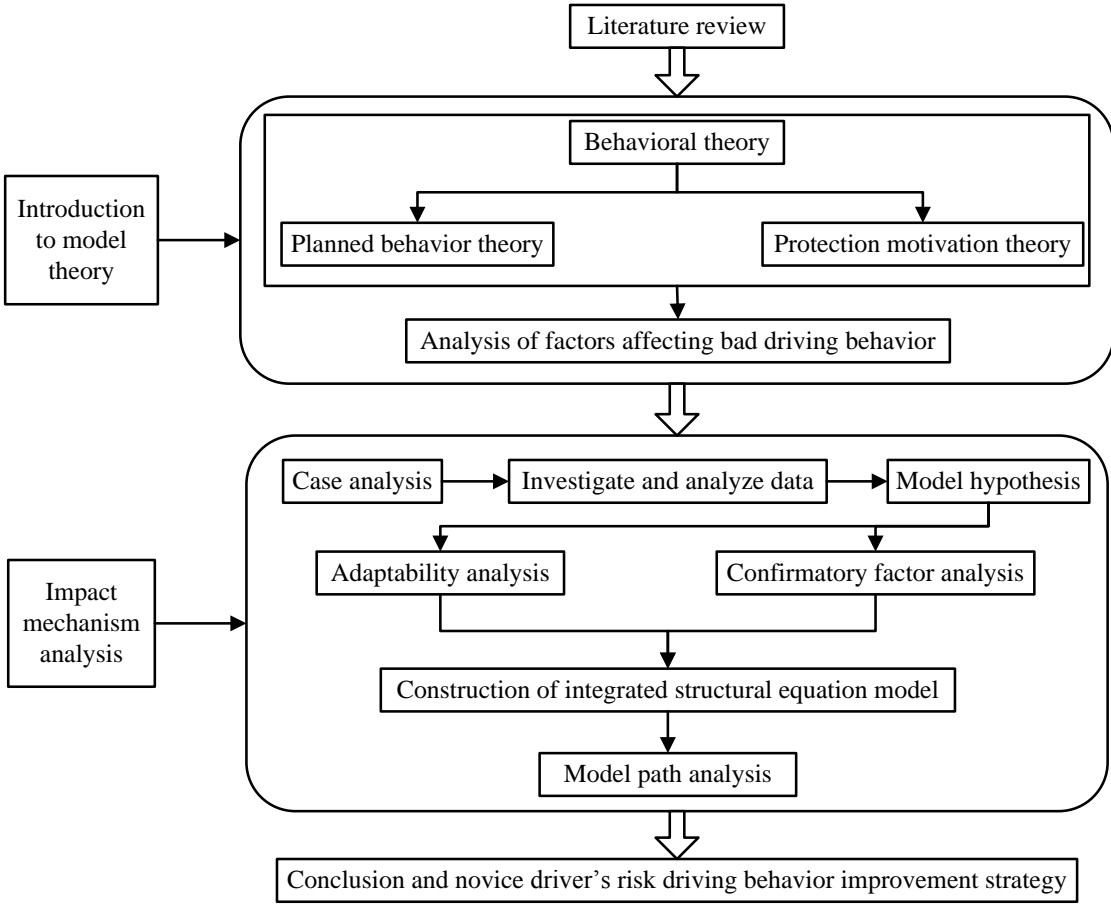

**Figure 1.** The overview of this paper.

## 2. Literature Review

The road traffic accidents caused by bad driving behavior are becoming increasingly more serious. Novice drivers are a high risk group in terms of bad driving behavior. Most traffic accidents caused by novice drivers are related to their dangerous driving behavior [1]. Generally, novice drivers pay greater attention to the less important details of the environment, and they often lack the holistic perception of the dangers in the traffic setting [2]. In contrast, experienced drivers have the overall perception of the traffic environment [3]. In addition, novice drivers are more likely to engage in risky driving behavior when compared with experienced drivers, and they are more aggressive when driving on the road [4–6]. Bad driving behavior is associated with less driving experience [7]. Day et al. discussed the high-risk factors of new drivers through interviews, and put forward suggestions to reduce the risky driving behavior of new drivers [8]. Ma et al. studied the causes of the high incidence of accidents among novice drivers, as well as the differences between novice drivers and experienced drivers in terms of risk perception [9]. However, the formation mechanism of the bad driving behavior of novice drivers and the analysis and quantification of the key factors leading to the bad driving behavior of novice drivers are seldom studied, which can be used to provide some theoretical basis for correcting the bad driving behavior and provide strategies and suggestions for

relevant departments and individuals. Salmon et al. collected and analyzed data on the factors that lead to drivers engaging in five fatal types of driving behavior. The results were mapped onto a system ergonomics model of the road transportation system in Queensland, Australia, to show where in the system the factors reside [10]. Xiang et al. constructed a data acquisition system by using an InvenSense's six-axis inertial measurement unit (IMU), and developed a fuzzy synthetical evaluation model combined with ISO (international organization for standardization) 2631-1:1997/Amd 1:2010 standard [11]. Sârbescu et al. aimed at exploring the intra-individual variation of dangerous driving behavior (errors, violations, and aggressive driving) and verified whether the outcomes could be predicted by both situational variables (weekly kilometers, felt traffic pressure, and traffic mood) and dispositional variables (Big-Five personality factors, age, and gender) [12].

Current research on dangerous driving behavior based on TPB theory and PMT theory mainly focuses on age, gender, driver behavior decision-making, vehicle types, and abnormal driving, among other factors. Research by Nancy Rhodes and Kelly Pivik shows that positive influence and risk perception are two factors of driver behavior decision-making. Importantly, positive influence contributes more to male and adolescent driving behavior than female and adult drivers [13]. Chung used the theory of planned behavior (TPB) to represent the intentional decision-making mechanism, and used the intensity of habits to reflect the intuitive decision-making process. Meanwhile, he applied the TPB model and AAR (the anticipated affective reactions) to study the influence of emotional arousal on speeding behavior [14]. Catherine Jolton, Mark Connor, and Samantha Jameson's research addresses this problem by studying the effects of variables on the intention to participate in three kinds of speed-related behaviors. A series of questionnaires based on TPB were used to assess the main motivation factors affecting motorcycle drivers' abnormal driving behavior, which was similar to those observed in the field of risky driving [15]. Castanier et al. used the theory of planned behavior to predict drivers' intentions for road violation [16]. On the basis of the theory of planned behavior, Li et al. developed the structural equation model of the relationship between competitive driving intention and behavior, and proposed an effective method to correct drivers' competitive driving behavior [17]. Kergoat studied the principle of various forms of speed limit information acting on overspeed driving behavior by using the protection motivation theory and deterrence theory, and found that under the condition of speed limit, the protection motivation theory has a better explanatory power of the acceleration intention of drivers [18]. Glendon and Walker studied the mechanism of the speed limit information on the overspeed behavior by using the protection motive theory [19]. Jiang et al. used the planned behavior theory to study drivers' fatigue driving behavior, proving that subjective norms, perceived behavior control, and orientation have significant influence on fatigue driving behavior [20]. Rowe et al. studied risky driving behaviors, such as speeding and distracted driving, by using the planned behavior theory, and found that attitude has the strongest explanatory power of all the risky driving behavior intentions, finding that the control of subjective norms and perceived behaviors on dangerous driving behaviors is also significant [21]. Chen introduced the concepts of green value perception, green useful sexy knowledge, and green comfort perception, exploring the factors influencing the loyalty of the public bicycle system by using the planning behavior theory, and found that green comfort perception and subjective norms have the strongest explanatory power for user loyalty [22]. Brijs et al. used the theory of planned behavior to explain and predict the behavior of drivers using mobile phones in the driving process, and put forward various measures to reduce the phenomenon of drivers using mobile phones in the driving process [23]. Eherenfreund et al. reasonably explained the dangerous driving behavior of young drivers by adding emotional factors to the influencing factors of the planned behavior theory, showing how the research results should be in the design of road safety information [24]. The theory of planned behavior and the theory of protective motivation can analyze bad driving behavior from different perspectives, study its formation mechanism, and formulate strategies from different perspectives, such as early education and late rewards and punishments, to improve the bad driving behavior of drivers. On the basis of TPB theory and PMT theory, this paper establishes a comprehensive structural

model. Through a case study, the proposed method can summarize the factors and mechanisms of the formation of bad driving behavior of novice drivers, providing a theoretical basis for future research on bad driving behavior.

By combing the existing research on motor vehicle drivers' bad driving behavior at home and abroad and the application of relevant behavioral theories, this paper summarizes the characteristics of the research and determines the direction of the paper.

(1) There is little research on the psychological factors of motor vehicle drivers' bad driving behaviors.

At present, bad driving behavior research is mainly based on accident data; scholars have analyzed and forecasted bad driving behavior from the objective conditions of traffic accidents. At the same time, motor vehicle drivers' subjective understanding and acceptance of bad driving behaviors largely affects the intention of their bad driving behavior, and a driver's bad driving behavior intention constitutes their behavioral attitude, subjective norms, perceived behavior control, the threat severity, internal and external return, response cost, the reaction efficiency, and the subjective effect of self-efficacy. Discovering how to quantify the drivers of the contribution of these psychological factors on bad driving behaviors and explaining the inner mechanisms of the formation of bad driving behavior will improve motor vehicle drivers' bad driving behavior, an important premise of road traffic safety.

(2) The comprehensive application of behavioral theory is rarely explored.

At present, there are more and more studies on the theory of planned behavior and the theory of protection motivation in the field of transportation. However, the explanation of planned behavior theory focuses on the early education and attitude change of the behavior subject. This method needs to improve the behavior of the behavior subject from at its root, requiring a long period of subtle education to achieve a better implementation effect. In comparison, the application of protection motivation theory in the field of transportation is not as good as that of planned behavior theory. At the same time, it needs to increase the influence of the consequences to make behavior the main body of accepting or rejecting this behavior; this method can establish relevant policy rigidity, intensifying the late penalty or reward to regulate behavior. In conclusion, the theory of planned behavior and the theory of protection motivation are able to analyze bad driving behavior from different perspectives; study its formation mechanism; and formulate strategies from different perspectives, such as early education and late rewards and punishments, to improve drivers' bad driving behavior.

## 3. Methodology

### 3.1. Theory of Planned Behavior

TPB is an extension of the rational behavior theory. It negates the hypothesis that the subject is a rational person. TPB associates a person's behavior with their beliefs. It no longer considers that the actual behavior of the subject is determined only by intention. It also holds that perceived behavioral control can shape a person's behavior. It holds that attitude, subjective norms, and perceived behavioral control are the three main variables affecting behavioral intention and behavior. The more positive their attitude, the greater the support of important others, and the stronger their perceived behavioral control is, the greater their behavioral intention. The three variables are independent of each other yet are related to each other. These three variables are influenced by personal behavioral beliefs, normative beliefs, and control beliefs, respectively.

TPB assumes that behavioral attitudes have a positive impact on the behavioral intentions of the actors. That is, the more positive the behavioral attitudes are, the stronger the willingness of the actors [25]. Perceived behavior control refers to the difficulty level of the subject in the process of perceiving the execution of a specific behavior. The model assumes that perceived behavior control directly affects the behavior intention and actual behavior of the actors, having a positive impact on both. That is to say, the more easily an agent can perceive the execution of an act, the stronger their intention to carry out the act and the greater the possibility of their actual execution of the action.

The comprehensive effects of attitude, subjective norms, and perceived behavior control are reasonable explanations for the actual behavior of the actors and have a great influence on behavior prediction.

When studying the influencing factors and the formation mechanism of the bad driving behavior of novice drivers, TPB theory can be used to analyze the factors affecting the actual driving behavior of novice drivers from many aspects. Unlike the theory of reasoned action (TRA), TPB theory has perceptual behavior control, which often reflects personal past experience, second-hand information, or expected obstacles. The more resources and opportunities for individuals they have, the less obstacles they expect, and the stronger their perceptual behavior control will be. TPB theory has been applied to the study of many bad driving behaviors in the field of transportation, which can better study the relationships between novice drivers' bad driving behavior and its subjective factors, social cognitive factors, and traffic education factors from a psychological point of view.

### 3.2. Protection Motivation Theory

PMT theory is the main theory of behavioral change, which explains behavioral change through threat assessment and response assessment in the cognitive regulation process. According to the health belief model, the adoption of a behavior depends mainly on its trust in short-term behavior. In 1983, Rogers put forward the theory of protective motivation, that is, the rewards factor is added to the health belief model, which considers the effect of "return" on behavior development in the long-term process. It can be said that PMT is an extension of the health belief model (HBM).

Compared with the health belief model, PMT pays more attention to the long-term cognitive process of behavior, and considers the influence of environmental factors and social factors on behavior. PMT has two important elements: one is internal reward, that is, the subjective pleasure of harmful behavior; the other is external reward, that is, the objective benefit of harmful behavior. According to the pattern of behavior formation, PMT is divided into three parts: information source, such as the environment or individuals; cognitive mediation process; and coping style. PMT theory explains the meaning of "fear appeal". It believes that threat information in the environment and in individuals will lead to individual threat assessment and response assessment. Threat assessment is the assessment of risky behavior. It is a comprehensive result of individual perception of threat severity, susceptibility, and return [26]. Coping ability evaluation is a comprehensive result of response effectiveness, self-efficacy, and response cost, which determines whether an individual has motivation to accept persuasion to avoid injury. Within the research background of this paper, on the basis of the theory of protective motivation, drivers' bad driving behavior can be explained in terms of their awareness of social and traffic policies. On this basis, we can standardize drivers' bad driving behavior by increasing penalties for bad driving behavior and making drivers aware of the serious consequences of bad driving behavior through simulation driving.

### 3.3. Classification of Bad Driving Behavior

The novice driver's bad driving behavior is complex and has complicated influencing factors. Therefore, studying bad driving behavior is very challenging. According to behavioral psychology, different behaviors are caused by different psychological processes. To comprehensively and effectively analyze bad driving behavior, it is necessary to classify bad driving behavior and thus reduce the complexity of risky driving behavior analysis. According to the different behavioral mental states of the driver's bad driving behavior, bad driving behavior can be divided into risky behavior, negligent behavior, and intentional violation behavior. The three types of risky driving behavior have different features.

(a) Risky behavior

Risky behavior refers to habitual risky driving behavior in which the driver fails to correct the wrong habits caused by lack of cognition in the early stage of driving experience, such as changing lanes at will, turning around at random, not wearing a seatbelt, not looking at the rearview mirror, driving on a line, and not using a turn signal.

(b)　Negligent behavior

Negligent behavior is divided into two types: operational errors and overconfidence due to negligence. Examples include the misuse of the throttle brake, improper steering, lane departure, driving close to other cars, overtaking at a corner, and misuse of the high beam.

(c)　Intentional violation behavior

Intentional violation behavior is a violation of the rules by intention. Examples are speeding, drunk driving, using a mobile phone while driving, retrograde behavior, illegally reversing, not following the rules, and illegal parking.

*3.4. Analysis of Factors Affecting Bad Driving Behavior*

The key factors affecting bad driving behavior of novice drivers were analyzed combined with the characteristics of bad driving behavior.

(a)　The attitude of bad driving behavior

The attitude of bad driving behavior is the affirmation or negation of bad driving behavior caused by the novice driver in the case of a priori basis and self-perception, pros and cons, and a close or alienated stable emotional state. The attitude of bad driving behavior is the direct feeling of the novice driver's bad driving behavior, which has a direct impact on the driving intention of bad driving behavior. The attitudes can be inferred through the individual's emotions (e.g., like and disgust, closeness and alienation) and rational perception (e.g., safety, benefit).

(b)　Accepting subjective norms of bad driving behavior

Accepting subjective norms of risky driving behavior is the behavioral motivation of a novice driver who wants their behavior to be consistent with social norms. The role of society in the behavior of individuals is significant and profound. In practice, accepting subjective norms of risky driving behavior can be measured by the influence of friends, family members, public advertising, and policies on behavioral intentions when novice drivers engage in bad driving behavior.

(c)　Perceived behavior control of bad driving behavior

Perceived behavior control of bad driving behavior is used to indicate the level of ability of a novice driver in carrying out bad driving behavior. Perceived behavioral control of bad driving behavior is measured by the novice driver's driving skills, conscious behavioral ability, and ability to bear economic risks.

(d)　Threat susceptibility of bad driving behavior

Threat susceptibility refers to the likelihood of an individual's perceived risk factors. The susceptibility towards bad driving behavior is related to the prevalence and susceptibility factors of bad driving behavior. If there are serious traffic accidents or serious fines caused by bad driving behavior around the subject, the subject will bring a sense of crisis and evasion awareness.

(e)　Threat severity of bad driving behavior

Threat severity refers to the degree of serious threat that a risk factor may pose to an individual's own interests. It is a subjective feeling. Therefore, different drivers have different judgments on the seriousness of risky driving behavior.

(f)　Bad driving behavior reward

Reward refers to the individual's self-satisfaction and external benefits due to individual behavior. The risky driving behavior reward refers to the pride and satisfaction of the novice driver after performing a risky driving behavior, as well as the convenience of time or distance in exchange for risky driving behavior.

(g)   Response cost of bad driving behavior

The response cost refers to the cost encountered by an individual when performing certain actions. When a novice driver engages in risky driving behavior, he may have to bear the psychological burden of traffic enforcement and the physical or linguistic condemnation of other drivers who comply with traffic rules.

(h)   Response effectiveness of bad driving behavior

Response effectiveness refers to an individual's awareness of whether a certain behavior is effective. Perceived behavioral control and self-efficacy operational definitions are not necessarily distinct. Novice drivers perform bad driving behaviors because they believe they can benefit from them. The target of response efficacy should be safe driving. Novice drivers can demonstrate their superior driving skills, avoid traffic jams, and avoid traffic charges when performing bad driving behavior. That is, the more the novice driver believes that such bad driving behavior is beneficial to them, the easier it is for them to perform such behavior.

(i)   Self-efficacy of bad driving behavior

Self-efficacy refers to the individual's perception of their ability to perform certain behaviors, that is, subjective feelings, beliefs, and judgments of the individual's behavior before completing a certain behavior. Novice drivers may feel that obeying traffic rules are a time-consuming, costly, and labor-intensive task. If there is no external force to supervise novice drivers' behavior, they may violate traffic rules. Conversely, if the novice driver feels that their driving skills are very high and that there is no possibility of accidents occurring when performing risky driving behavior, the driver may maximize their risky driving behavior.

## 4. Case Analysis

Firstly, the RP (revealed preference survey)/SP (stated preference survey) survey method was used to collect basic personal information of drivers, the cognitive status of bad driving, and the corresponding characterization information about bad driving behavior. Then, the descriptive statistical analysis and reliability and validity test of the survey data were carried out by using the method of mathematical statistics. Characteristics of drivers' bad driving behavior were summarized, laying out a foundation for the establishment and fitting of the integrated model of drivers' bad driving behavior.

### 4.1. Design of Survey Questionnaire for Bad Drivers' Driving Behavior

Many studies have shown that drivers with driving experience of less than or equal to three years generally have poor driving skills, have poor physical and mental quality, easily underestimate the risk of road traffic, and lack good vehicle driving predictability [27–30]. Moreover, driving experience is also related to driving mileage. According to relevant research and surveys, the annual driving mileage of Chinese non-professional motor vehicle drivers ranges from 10,000 to 15,000 km —the lower limit being 10,000 km. Thus, a novice driver in this paper is defined as a driver with driving experience of less than 3 years and a driving mileage of less than 30,000 km.

We used a simple random sampling survey to collect specific personal attributes and information on bad driving behavior from Chinese novice drivers through online responses and roadside inquiry. We collected a total of 313 questionnaires and deleted 29 invalid questionnaires with incomplete answers and multiple extreme answers. A total of 284 valid questionnaires were collected, and the effective response rate of this questionnaire reached 90.73%. The sample capacity requirements were met.

The questionnaire was divided into three parts. The first part focused on descriptive statistical information, such as the personal attributes of the respondents. The second part investigated the characterization information of risky driving behavior. The third part investigated the cognitive status of risky driving behavior.

Descriptive statistics represent individual differences in motor vehicle drivers' bad driving behavior. It is necessary to establish a direct link between bad driving behavior and descriptive statistical data in order to facilitate a targeted analysis of bad driving behavior of different samples. The RP survey method was used to obtain the actual situation of the respondent. The descriptive statistical variables in this paper include the age, gender, driving experience and driving mileage, and degree of education, which correspond to the design of four questions, as shown in Table 1.

**Table 1.** Descriptive statistics survey.

| According to your actual situation, write √ on the corresponding options |
| :---: |

| | |
| :---: | :---: |
| I Your gender is: | |
| □ Male | □ Female |
| II Your age is: | |
| □ Between 18 and 28 years old (including 18 and 28 years old) | □ Over 28 years old |
| III Your driving experience and driving mileage are: | |
| □ Driving age ≤3 years or driving mileage ≤30,000 km | |
| □ Driving age 3 years and driving mileage >30,000 km | |
| IV Your education is: | |
| □ Elementary school and below | □ Junior high school |
| □ High school/college | □ Bachelor's degree or above |

The bad driving behavior characterization information mainly presented the specific risky driving behavior of the respondents in their daily life, which also reflects the degree of concern of the motor vehicle driver towards risky driving behavior. The RP survey method was used to determine the actual experience of the respondent. The survey used a Likert-style scale to score five points; that is, on the five semantic difference scales, the three types of typical violations were evaluated. Among them, risky behavior was characterized by eight issues: ITEM1, ITEM2, ITEM3, ITEM4, ITEM5, ITEM6, ITEM7, and ITEM8. Negligent behavior was characterized by five issues: ITEM9, ITEM10, ITEM11, ITEM12, and ITEM13. Intentional violations were characterized by six issues: ITEM14, ITEM15, ITEM16, ITEM17, ITEM18, and ITEM19. Finally, the respondent chose the option of 'agree/disagree'. The higher the score, the higher the enthusiasm for this kind of risky driving behavior. A total of 19 questions were designed for the questionnaire, as shown in Table 2.

**Table 2.** Survey of risky driving behavior characterization information.

| Variates | Numbers | Questions |
| :--- | :--- | :--- |
| Risky behavior | ITEM1 | During the driving process, I will change lanes according to my own needs and mood. |
| | ITEM2 | During the driving process, I will turn around according to my own needs and mood. |
| | ITEM3 | During the driving process, I sometimes suddenly find myself forgetting to wear a seatbelt. |
| | ITEM4 | During the driving process, I am not used to looking at the rearview mirror. |
| | ITEM5 | During the driving process, I sometimes suddenly drive the car onto the solid line. |
| | ITEM6 | During the driving process, I often forget to use the turn signal. |
| | ITEM7 | I may choose to overtake because of time pressure or other reasons, sometimes even when driving in curved lanes. |
| | ITEM8 | I always follow the preceding vehicle closely to stop neighboring vehicles from queue-jumping. |
| Negligent behaviors | ITEM9 | I always confuse the throttle and the brake in an emergency. |
| | ITEM10 | I always need to adjust the steering wheel several times to get the car in the right position. |
| | ITEM11 | I always find myself not driving in the center of the lane. |
| | ITEM12 | I always misuse high beams when passing other vehicles. |
| | ITEM13 | I always find my car is speeding when looking at the speedometer. |
| Intentional behaviors | ITEM14 | When I am confident to drive, I will choose to drive by myself, even if I have drunk a certain amount of alcohol. |
| | ITEM15 | When there is a phone call, I will choose to check my mobile for fear of delaying my work. |
| | ITEM16 | Because of time pressure or other reasons, I will sometimes choose to reverse drive. |
| | ITEM17 | Sometimes when road conditions do not allow parking, but I have an urgent demand, I will choose to park my car. |
| | ITEM18 | Sometimes, when there are no parking lots around restaurants or my workplace, I will choose to park on the roadside. |
| | ITEM19 | When there are no cars and pedestrians at intersections, I will drive past the intersections in spite of a red light. |

The survey of the cognitive status of bad driving behaviors was the main part of the questionnaire. The results were used to explain the mechanism of the formation of risky driving behaviors. Here, the RP method and SP method were used. First, the RP method can be used to determine the actual experience and preferences of the drivers. Then, the SP method is used to obtain the intentional preferences of the driver. Semantic differentials in the five scores are also used for the collection of such data. Nine latent variables in five semantic differential scales were considered: attitudes towards bad driving behaviors, subjective criteria, threat sensitivity, threat severity, rewards, response cost, response efficiency, self-efficiency, and behavior intentions. Each latent variable was characterized by three observed variables. Finally, respondents were asked to choose among some 'agree/disagree' options. A higher score shows a higher probability to engage in risky driving behaviors. Thirty-one questions were included in the questionnaire, as shown in Table 3.

**Table 3.** Survey of the cognitive status of the bad driving behaviors.

| Latent variables | Items | Questions |
|---|---|---|
| Attitudes towards bad driving behaviors | ITEM20 | I hate risky driving. I always drive normally. |
| | ITEM21 | I think sometimes I do not care about the appeal of safe driving. |
| | ITEM22 | Some traffic regulations or regulations are unreasonable and need not be strictly observed. |
| | ITEM23 | My friends and relatives think that I can engage in risky driving behaviors properly. |
| Subjective criteria | ITEM24 | I think my friends and relatives will worry about me if I engage in risky driving behaviors. |
| | ITEM25 | My friends and relatives may engage in risky driving behaviors in their daily life. |
| | ITEM26 | I think risky driving behaviors will be noticed by traffic policemen or video cameras. |
| Threat sensitivity | ITEM27 | I think my risky driving behaviors will be easily noticed by law enforcers. |
| | ITEM28 | When I engage in risky driving, it is within my control, just like safe driving. |
| | ITEM29 | I think risky driving behavior can cause serious traffic accidents. |
| Threat severity | ITEM30 | I don't think risky driving behaviors will be severe enough to cause traffic accidents. |
| | ITEM31 | I think engaging in risky driving behaviors will disrupt my life to a certain degree. |
| | ITEM32 | I think engaging in risky driving behaviors will save lots of time. |
| Rewards | ITEM33 | I think risky driving behaviors can bring mental pleasure for me. |
| | ITEM34 | I think engaging in risky driving behaviors can shorten the trip distance. |
| | ITEM35 | I think engaging in risky driving behaviors may cause traffic chaos. |
| Response cost | ITEM36 | I think if I engage in dangerous driving behaviors, it will bring trouble to other travelers. |
| | ITEM37 | I think drivers around will sound their car horns or blame me if I engage in risky driving behaviors. |
| | ITEM38 | I think engaging in risky driving behaviors can improve my traffic efficiency. |
| Response efficiency | ITEM39 | I think engaging in risky driving behaviors can reduce the time lost in congestion. |
| | ITEM40 | I think engaging in risky driving behaviors can avoid some traffic fees. |
| | ITEM41 | I am confident to engage in risky driving behaviors if I am decided. |
| Self-efficiency | ITEM42 | I am confident that I can keep myself safe when engaging in risky driving behaviors. |
| | ITEM43 | I can afford the fine for risky driving behaviors. |
| | ITEM44 | I always try to engage in risky driving behaviors when driving. |
| Behavioral intentions | ITEM45 | Sometimes I have to engage in risky driving behaviors. |
| | ITEM46 | I predict that I will certainly engage in risky driving behaviors. |

## 4.2. Descriptive Statistical Analysis

Descriptive statistical analysis is an important basis to guarantee the generality, representativeness, and reliability of an investigation sample. SPSS software was used to analyze the data obtained from the questionnaires, which showed the distribution of gender, age, driving experience, driving mileage, and educational level of the drivers. Thereby, we could determine the descriptive statistical characteristics of the sample and do our best to make the sample meet the requirements of the respondents. Take a certain age interval as an example, in order to avoid errors caused by a simple sample, distribution of various characteristics should be uniform, and statistics of each variable should be analyzed.

(1) We compiled descriptive statistics about the personal characteristics of drivers. The results are shown in Table 4.

**Table 4.** Descriptive statistics of the personal characteristics of drivers.

| Items | Options | Frequency | Rate |
|---|---|---|---|
| Gender | Male | 150 | 52.82% |
| | Female | 134 | 47.18% |
| Age | (18,28) | 182 | 64.08% |
| | (28,70) | 102 | 35.92% |
| Driving experience and driving mileage | Driving experience <3 years or mileage <30,000 km | 215 | 75.7% |
| | Driving experience >3 years or mileage >30,000 km | 69 | 24.3% |
| Educational level | Primary school or below | 3 | 1.06% |
| | Junior high school | 56 | 19.72% |
| | High school or specialist | 60 | 21.13% |
| | Bachelor or above | 165 | 58.1% |

In the survey, males were more willing to cooperate and were more likely to drive in the family; therefore, the rate of males was slightly higher. Most drivers were younger than 28 years old. The educational level of the respondents was generally at the high school level or above. The driving experience of respondents was mostly 3 years or less, while the mileage was mostly 30,000 km or less. Both of these factors have a certain degree of consistency, which is related to the boom of the Chinese driver's license test.

(2) We compiled descriptive statistics of the observed variables of the risky driving behaviors. The results are shown in Table 5.

**Table 5.** Observed variables of the risky driving behaviors.

| Kinds | Items | Subjects | Minimum value | Maximum value | Mean value | Standard error | Standard deviation |
|---|---|---|---|---|---|---|---|
| Bad behaviors | ITEM1 | Change lanes randomly | 1 | 5 | 2.47 | 1.362 | 0.081 |
| | ITEM2 | Make U-turns randomly | 1 | 5 | 2.04 | 1.221 | 0.072 |
| | ITEM3 | Not fastening seatbelts | 1 | 5 | 2.24 | 1.201 | 0.071 |
| | ITEM4 | Not watching rearview mirrors | 1 | 5 | 1.87 | 1.075 | 0.064 |
| | ITEM5 | Drive along center line | 1 | 5 | 1.87 | 1.080 | 0.064 |
| Negligent behaviors | ITEM6 | Not using steering lamp | 1 | 5 | 1.75 | 1.039 | 0.062 |
| | ITEM7 | Confusing throttle and brake | 1 | 5 | 1.74 | 1.004 | 0.060 |
| | ITEM8 | Improper steering | 1 | 5 | 2.05 | 1.145 | 0.068 |
| | ITEM9 | Driving across center line | 1 | 5 | 1.77 | 1.081 | 0.064 |
| | ITEM10 | Overtaking on a curve | 1 | 5 | 2.52 | 1.220 | 0.072 |
| | ITEM11 | Misusing high beams | 1 | 5 | 2.15 | 1.133 | 0.067 |
| | ITEM12 | Following vehicles closely | 1 | 5 | 1.79 | 0.950 | 0.056 |
| Intentional behaviors | ITEM13 | Speeding | 1 | 5 | 1.89 | 1.034 | 0.061 |
| | ITEM14 | Driving drunk | 1 | 5 | 1.35 | 0.828 | 0.049 |
| | ITEM15 | Using mobile phone when driving | 1 | 5 | 1.85 | 1.033 | 0.061 |
| | ITEM16 | Driving inversely | 1 | 5 | 1.37 | 0.816 | 0.048 |
| | ITEM17 | Parking illegally | 1 | 5 | 1.93 | 1.076 | 0.064 |
| | ITEM18 | Berthing illegally | 1 | 5 | 2.20 | 1.088 | 0.065 |
| | ITEM19 | Going against rules | 1 | 5 | 1.42 | 0.851 | 0.051 |

According to Table 5, changing lanes randomly and not fastening seat belts in the bad behaviors, overtaking in curve in the negligent behaviors, and berthing illegally in the intentional behaviors were more common. These bad and negligent behaviors generally indicate that Chinese driving training institutions must have stricter training mechanisms. In addition to good driving skills, drivers should also be trained in common knowledge and driving habits. These intentional behaviors not only indicate the lack of knowledge of drivers of traffic laws and regulations, but also reflect the attitude of drivers in dealing with irregularities and the weak deterrent effect of law enforcement on drivers' behaviors in China.

(3) After sorting out the data by SPSS software, the Cronbach's alpha coefficients of each variable and its corresponding dimensions were analyzed. The coefficient of internal consistency is shown in Table 6 below.

**Table 6.** Coefficient of internal consistency.

|  | Number of Items | Cronbach's alpha | Standard | Result |
|---|---|---|---|---|
| Bad behaviors | 4 | 0.764 | >0.6 | ok |
| Negligent behaviors | 4 | 0.752 | >0.6 | ok |
| Intentional behaviors | 7 | 0.871 | >0.6 | ok |
| Attitudes towards behaviors | 3 | 0.631 | >0.6 | ok |
| Subjective norm | 3 | 0.668 | >0.6 | ok |
| Threat sensitivity | 3 | 0.733 | >0.6 | ok |
| Threat severity | 3 | 0.694 | >0.6 | ok |
| Rewards | 3 | 0.846 | >0.6 | ok |
| Response cost | 3 | 0.831 | >0.6 | ok |
| Response efficiency | 3 | 0.822 | >0.6 | ok |
| Self-efficiency | 3 | 0.609 | >0.6 | ok |
| Behavioral intention | 3 | 0.712 | >0.6 | ok |

From Table 6, it can be seen that the internal consistency coefficients of each scale and its dimensions met the test requirements, and that the test reliability was good.

## 5. Model Establishment

The TPA and PMT have similarities but also different focuses and uniqueness. Therefore, the TPB and PMT theories are combined and complement each other, causing the degree of interpretation to also improve [31]. The self-efficiency parts are similar in the two theories; thus, they can be combined as a cross-item. In summary, we integrated the two theories to lay out the path hypothesis below and establish the initial integrated structural equation model of the two theories. The structure of the model is shown in Figure 2.

H0: Bad driving behavior of novice drivers is influenced by threat sensitivity, threat severity, rewards, response cost, response efficiency, self-efficiency, subjective norms, and attitudes towards behaviors (above the line).

H1: Threat sensitivity of the drivers has a negative effect on risky driving behavior intentions.

H2: Threat severity of the drivers has a negative effect on risky driving behavior intentions.

H3: Rewards of the drivers have a positive effect on risky driving behavior intentions.

H4: Response cost of the drivers has a negative effect on risky driving behavior intentions.

H5: Response efficiency of the drivers has a negative effect on risky driving behavior intentions.

H6: Self-efficiency of the drivers has a positive effect on risky driving behavior intentions.

H7: Subjective norm of the drivers has a positive effect on risky driving behavior intentions.

H8: Attitudes towards behaviors of the drivers have a positive effect on risky driving behavior intentions.

H9: Risky driving behavior intentions of the drivers have a positive effect on bad behaviors.

H10: Risky driving behavior intentions of the drivers have a positive effect on negligent behaviors.

H11: Risky driving behavior intentions of the drivers have a positive effect on intentional behaviors.
H12: Self-efficiency of the drivers has a positive effect on bad behaviors.
H13: Self-efficiency of the drivers has a positive effect on negligent behaviors.
H14: Self-efficiency of the drivers has a positive effect on intentional behaviors.

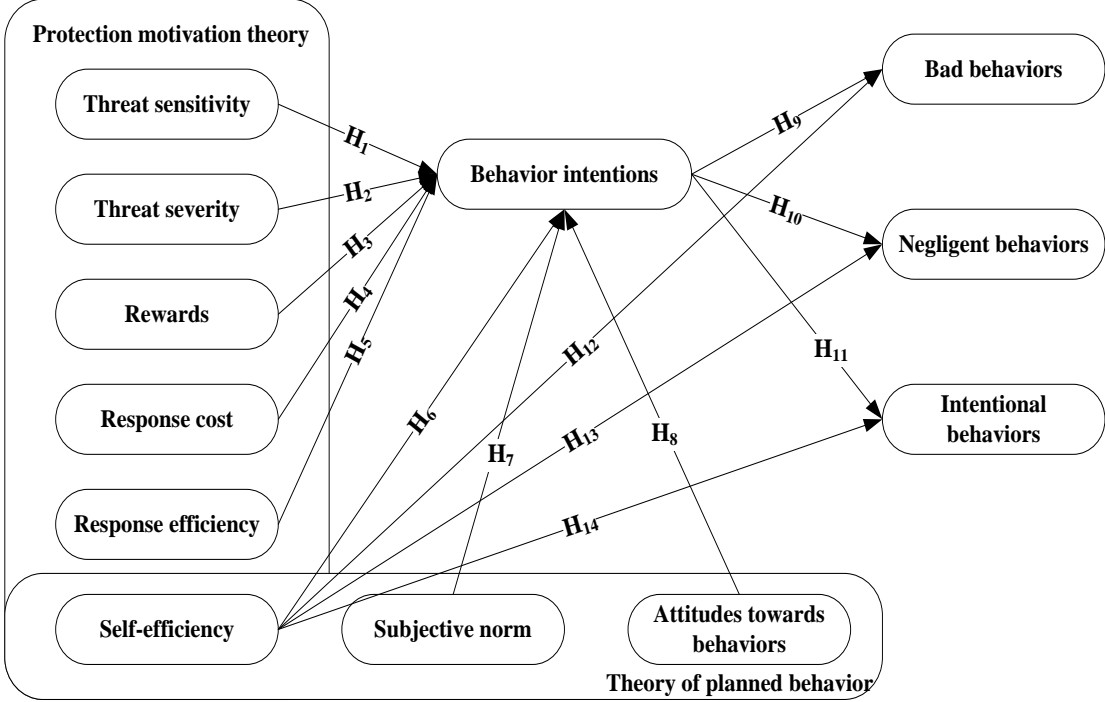

**Figure 2.** The integrated model of risky driving behaviors.

## 5.1. Evaluation of Fitness

We used the AMOS (advanced mortar system) to estimate the fit parameters of the novice drivers' bad driving behavior model. AMOS is a software that uses structural equations to explore relationships between variables. It contains analysis of variance, covariance, hypothesis testing, and other basic analysis methods. The results are shown in Table 7. The fitting result showed that the values of each fitting evaluation index of the model were all located at a good point in the interval of critical values, which shows that the overall fit of the model was ideal and that the model had good predictive ability. The latent variable had good predictive ability for the observed variables.

**Table 7.** Fitness parameters of the AMOS (advanced mortar system) model.

| Fitting Evaluation Indexes | Index Values | Critical Values |
| --- | --- | --- |
| Standardized root mean square residual (SRMR) | 0.215 | <0.50 |
| Comparative fit index (CFI) | 0.979 | >0.90 |
| Root mean square error approximate (RMSEA) | 0.064 | <0.08 |
| Normed fit index (NFI) | 0.935 | >0.90 |
| Goodness of fit index (GFI) | 0.924 | >0.90 |
| Adjustment of goodness of fit index (AGFI) | 0.937 | >0.90 |
| Chi-square dof ratio ($\chi^2$/df) | 2.970 | <5.00 |

## 5.2. Confirmatory Factor Analysis

The load factors showed the importance of the observed variables on the latent variables and were also important indicators for evaluating the basic fit of the model. The results of confirmatory

factor analysis of the model using AMOS are shown in Table 8. The results showed that all the load factors were greater than 0.5, and that the model had a good fit.

**Table 8.** The results of the confirmatory factor analysis of the risky driving behaviors of the novice drivers.

| Paths of the Measurement Model | | | Load Factors | *T* Values |
|---|---|---|---|---|
| ITEM1 | <- | Bad behaviors | 0.50 *** | 14.65 |
| ITEM2 | <- | Bad behaviors | 0.50 *** | 14.71 |
| ITEM5 | <- | Bad behaviors | 0.74 *** | 18.40 |
| ITEM6 | <- | Bad behaviors | 0.77 *** | 18.69 |
| ITEM8 | <- | Negligent behaviors | 0.66 *** | 16.78 |
| ITEM9 | <- | Negligent behaviors | 0.76 *** | 18.54 |
| ITEM10 | <- | Negligent behaviors | 0.54 *** | 14.76 |
| ITEM12 | <- | Negligent behaviors | 0.64 *** | 16.53 |
| ITEM13 | <- | Intentional behaviors | 0.61 *** | 16.27 |
| ITEM14 | <- | Intentional behaviors | 0.68 *** | 17.01 |
| ITEM15 | <- | Intentional behaviors | 0.66 *** | 16.72 |
| ITEM16 | <- | Intentional behaviors | 0.78 *** | 18.76 |
| ITEM17 | <- | Intentional behaviors | 0.64 *** | 16.55 |
| ITEM18 | <- | Intentional behaviors | 0.64 *** | 16.57 |
| ITEM19 | <- | Intentional behaviors | 0.87 *** | 34.98 |
| ITEM20 | <- | Attitudes towards behaviors | 0.56 *** | 14.83 |
| ITEM21 | <- | Attitudes towards behaviors | 0.54 *** | 14.74 |
| ITEM22 | <- | Attitudes towards behaviors | 0.63 *** | 16.47 |
| ITEM23 | <- | Subjective norm | 0.54 *** | 14.75 |
| ITEM24 | <- | Subjective norm | 0.55 *** | 14.79 |
| ITEM25 | <- | Subjective norm | 0.65 *** | 16.60 |
| ITEM26 | <- | Threat sensitivity | 0.62 *** | 16.31 |
| ITEM27 | <- | Threat sensitivity | 0.73 *** | 18.35 |
| ITEM28 | <- | Threat sensitivity | 0.54 *** | 14.75 |
| ITEM29 | <- | Threat severity | 0.72 *** | 18.29 |
| ITEM30 | <- | Threat severity | 0.56 *** | 14.85 |
| ITEM31 | <- | Threat severity | 0.66 *** | 16.73 |
| ITEM32 | <- | Rewards | 0.75 *** | 18.52 |
| ITEM33 | <- | Rewards | 0.82 *** | 31.77 |
| ITEM34 | <- | Rewards | 0.79 *** | 28.63 |
| ITEM35 | <- | Response cost | 0.84 *** | 32.16 |
| ITEM36 | <- | Response cost | 0.93 *** | 38.53 |
| ITEM37 | <- | Response cost | 0.64 *** | 16.56 |
| ITEM38 | <- | Response efficiency | 0.77 *** | 18.68 |
| ITEM39 | <- | Response efficiency | 0.70 *** | 18.25 |
| ITEM40 | <- | Response efficiency | 0.79 *** | 27.42 |
| ITEM41 | <- | Self-efficiency | 0.81 *** | 31.25 |
| ITEM42 | <- | Self-efficiency | 0.76 *** | 18.53 |
| ITEM43 | <- | Self-efficiency | 0.68 *** | 17.05 |
| ITEM44 | <- | Behavioral intention | 0.77 *** | 18.70 |
| ITEM45 | <- | Behavioral intention | 0.55 *** | 14.83 |
| ITEM46 | <- | Behavioral intention | 0.57 *** | 14.96 |

Note: *** = $p < 0.001$.

Because the scale was used to measure the same group of subjects at the same time, subjecting it to the influence of the social approval effect, the subjects may have had a tendency to answer when they answered. In order to control the deviation caused by this tendency, Harman's single factor analysis was used to analyze the data obtained by exploratory factor analysis. The results showed that nine common factors greater than one were extracted from all the questions. The first factor explained 33.612% of the total variation, reaching 40% of the test criteria, and thus there was no common method deviation in this study. The results of exploratory factor analysis were as follows: factor analysis could

only be carried out when the KMO (kaiser-meyer-olkin) test and battery ball test were carried out, KMO > 0.50 (significant probability of battery ball test statistics), and $p < 0.05$. In this study, KMO = 0.906, = 2122.952, $p = 0.000$ for the results of the initial test. It was suitable for factor analysis of the test results.

### 5.3. Path Analysis of Structural Equation Model

After the model confirmatory factor analysis was completed, the path analysis was carried out on the bad driving behavior of the novice drivers, and the path coefficient between the latent variables was obtained, as shown in Figure 3.

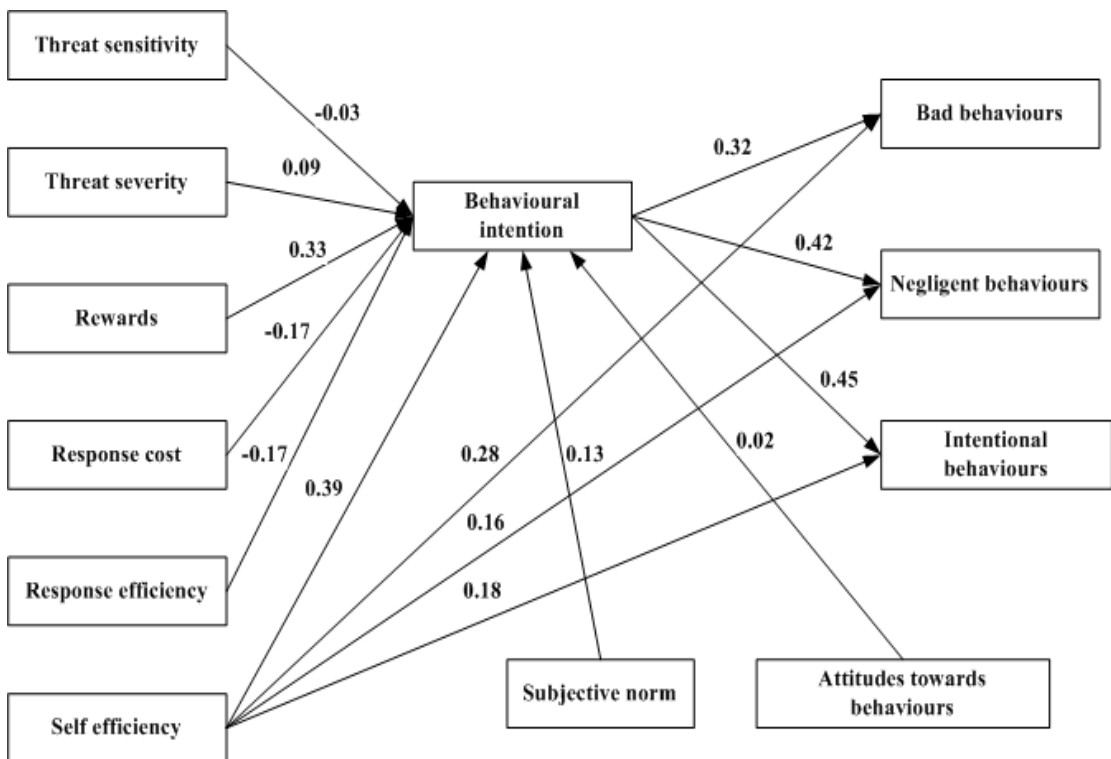

**Figure 3.** Integration model path of novice drivers' bad driving behavior.

(1) Explanatory power for bad driving behavioral intention from the structural model:

Bad driving behavioral intention of novice drivers = (−0.03) threat susceptibility +(0.09) threat severity +(0.33) rewards +(−0.17) response cost +(0.17) response efficiency +(0.39) self-efficiency +(0.02) behavior attitude +(0.13) subjective norms.

It was found that for the explanatory power of bad driving behavioral intention, reward and self-efficiency had the strongest explanatory power for novice drivers' behavioral intention. Self-efficiency had the strongest explanatory power for skilled drivers, and the other latent variables had significant influence on their behavioral intention. At the same time, the explanatory power of the other latent variables was comparable.

Because of the lack of driving experience, novice drivers have difficulty perceiving the threat of bad driving behavior, and their perception of threat susceptibility is low. Generally, novice drivers seldom experience or witness traffic accidents or social hazards caused by bad driving behavior. It is easy for them to underestimate the risk of bad driving behavior. As a result, they lack perceptual knowledge of the serious consequences of bad driving. They do not sufficiently pay attention to bad driving behavior and are less aware of the perception of threat severity. This also leads to a lack of knowledge about the cost of novice drivers' bad driving behavior. At the same time, since the cost

pressure of bad driving behavior is not high at present, the novice driver thinks that they can afford it, and their cost perception is low. Because the punishment mechanism for bad driving behavior and the reward mechanism for obeying traffic rules are perfect at present, the perception of social and individual interests brought about by obeying traffic rules is high; that is, the response efficiency perception of bad driving behavior is low. Generally, novice drivers are younger and have had less time to experience driving, which causes their ambiguous attitudes towards all kinds of driving behaviors as well as their conformity behavior. Their attitude towards bad driving behavior is also lower. At the same time, novice drivers more easily rebel, and there is a certain antagonistic mentality towards others' discouragement and demands. Therefore, their subjective norm has a negative impact on bad driving behavior intention, and their perception degree is high. Relatively speaking, novice drivers' bad driving behavior intention has a high perception of their reward. The convenience of time and space brought about by bad driving, as well as psychological satisfaction and mental stimulation, are temptations that novice drivers cannot resist. Self-efficacy is the most explanatory factor for novice drivers' bad driving behavior, which is closely related to their behavior to overestimate their driving skills and to underestimate the risk, and their lack of overall perception of the traffic environment leads to bad driving behavior.

(2) Explanatory power for bad habits from structural model:

Novice drivers' bad habits = (0.34) behavioral intention +(0.28) self-efficiency.

It was found that self-efficacy had strong explanatory power with regards to novice drivers' bad habits.

In terms of bad habits, such as random lane change, random turnaround, and not looking at rearview mirrors, novice drivers think more of self-efficacy. At the same time, behavioral intention also plays a role. This shows that novice drivers are timid and need to consider all factors. If the surrounding facilities or specific road conditions limit the self-efficiency of novice drivers, it may hurt their confidence in carrying out bad habits.

(3) Explanatory power for negligence from the structural model:

Novice drivers' negligence = (0.43) behavioral intention +(0.16) response efficiency.

It was found that the explanatory power of self-efficiency was more powerful for explaining the negligent behavior of novice drivers.

For negligence, such as overtaking on a curve and approaching cars, novice drivers consider more of their driving skills. When performing these behaviors, novice drivers may give up because they are not confident in their driving skills. For improper steering, lane departure, and other phenomena, the novice driver, who is unskilled, may not be able to reach the standard of normal driving behavior, even if they are sufficiently self-efficient, resulting in negligence.

(4) Explanatory power for deliberate irregularities from the structural model:

Novice drivers' deliberate irregularities = (0.46) behavioral intention +(0.18) self-efficiency.

It was found that self-efficiency and behavioral intention both have strong explanatory power for novice drivers' deliberate irregularities.

For novice drivers, self-efficiency is more influential than behavioral intention, which shows that novice drivers are more concerned about whether their driving skills can support their successful completion of deliberate irregularities. In some cases, if the novice drivers are not confident in their driving skills, they may choose against deliberate irregularities.

## 6. Improvement Strategy of Novice Drivers' Bad Driving Behavior

Through the above analysis, the causes and the specific mechanisms of the bad driving behavior of novice drivers have been made clear. This study shows that the model can better explain the causes and

the trigger mechanisms of bad driving behavior of novice drivers. Therefore, it is necessary to make targeted improvement strategies according to the characteristics of novice drivers' bad driving behavior.

(1) Establishing an advanced traffic monitoring system to reduce the susceptibility of novice drivers to bad driving risks.

It is necessary to reduce novice drivers' susceptibility to bad driving behavior. For this purpose, we should establish an advanced traffic monitoring system; use modern traffic monitoring equipment such as UAV(unmanned aerial vehicle), electronic police, and other modern traffic monitoring equipment; set them on the key sections and intersections; and constantly report road conditions to the central traffic monitoring system. If there is bad driving behavior of automatic vehicle drivers, it should be confirmed immediately so that bad driving behavior cannot escape punishment. At the same time, it is necessary to vigorously publicize that bad driving behavior will have a serious adverse impact on the surrounding traffic environment and traffic safety, that it can easily face police control, and that it will leave a bad impression on others, hurting the novice driver's own social image. Thus, the occurrence of bad driving behavior of novice drivers will be reduced, and the perception of bad driving behavior of novice drivers will be enhanced.

(2) Constructing the accident experience education and evaluation mechanism platform and enhancing the threat severity of drivers' bad driving behavior.

It is necessary to enhance novice drivers' perception of the threat severity of bad driving behavior. To increase education about accidents, we can carry out simulated hazard awareness training and strengthen the awareness of the risks of driving and bad driving behavior. At the same time, a comprehensive evaluation mechanism for motor vehicle driving behavior should be established, and the driving behavior of motor vehicle drivers should be strictly monitored at irregular intervals. Meanwhile, the results of the investigation should be put on record. If the cumulative number of bad driving behaviors reaches a certain level, the driver will be warned. For impenitent recidivism, it is necessary to organize regular study, education, and examination. Furthermore, undesirable drivers who have caused serious consequences many times should be punished directly by revoking their driver's license so that the novice driver fully understands the seriousness of bad driving behavior.

(3) Improving road design, enhancing simulated driving, and reducing novice drivers' bad driving rewards.

Novice drivers' perception of rewards for bad driving behavior should be reduced. First, in the road design stage, the constructor should focus on the rationality of the design. Driving schools need to increase the driver's accident experience in the simulated training stage for the novice driver to resist bad driving behavior, also eliminating the mental pleasure derived from bad driving behavior, such as by speeding. In this way, we can effectively correct the driver's bad driving behavior through early education.

(4) Establishing an online credit rating system to increase the cost of bad driving response for novice drivers.

It is necessary to enhance novice drivers' perception of the response cost for bad driving behaviors. For this purpose, we should set up an online evaluation system for evaluating the credit degree of motor vehicle drivers, which directly links the bad driving records of motor vehicle drivers with their credit degrees. Motor vehicle drivers' bad driving behavior records will directly affect their evaluation of work units, bank loans, and family credit to prevent bad driving. For bad driving behavior, such as drunk driving, road rage, and illegal driving, we need to increase the punishment period to effectively curb bad driving behavior and increase the response cost of such behavior.

(5) Constructing a reward and punishment mechanism for driving behavior to reduce the bad driving response efficiency of novice drivers.

To reduce novice drivers' perception of the response efficiency for bad driving behavior, we should establish a reward and punishment mechanism for bad driving behavior. On the one hand, motorists who have excellent performance and have accumulated zero points on the bad driving record in one year will be awarded the incentive to extend the audit period of their driver's license to enhance their

perception of the response efficiency for obeying traffic rules. At the same time, for motorists who show bad performance and accumulate more than six points on the bad driving record in one year, there will be a punishment of shortening the audit period of the driver's license. In the case of serious circumstances, the driver will not be granted a driver's license for the rest of his life. In this way, the perception of the response efficiency for bad driving behavior will be reduced.

(6) Developing driving training institutions to reduce the bad driving self-efficacy of novice drivers.

To promote novice drivers' correct understanding of their self-efficiency, we should increase the safe driving experience of novice drivers in driving school training institutions and fully and effectively regulate the driving habits of new drivers. In addition to learning driving skills, it is necessary to strengthen the learning of traffic rules and carry out simulated environment practice for the novice drivers to understand traffic rules more comprehensively, understand their driving skills and driving risks objectively, and lose their confidence in engaging in bad driving behavior.

(7) Being strict in driving school training and in the examination mechanism to correct novice drivers' attitudes towards bad driving behavior.

We should strengthen the education of novice drivers' attitudes towards bad driving behavior by taking advantage of their behavior and attitude, which can be changed easily. Most of China's traffic laws and regulations are learned in driving school; thus, the school must be responsible for the society and the whole traffic system, providing a good education on traffic laws and regulations by increasing the difficulty of the examination. It is necessary to focus on correcting the attitude of the driver's driving behavior, making them obey the traffic rules consciously. Therefore, the driving school and other training institutions need a strict training and assessment mechanism.

(8) Changing the educational form and constructing a subjective norm of bad driving for novice drivers.

We should take advantage of subjective norms' unique negative correlation impact effects on novice drivers. It shows that in practice, family and friends should change the educational form of bad driving behavior for novice drivers, as it is easy for novice drivers to produce adverse psychology. More encouraging and easily accepted language should be used to educate novice drivers. At the same time, family and friends should serve as examples by obeying traffic rules. In this way, there will be an imperceptible influence on novice drivers. This can reduce the bad driving behavior of novice drivers.

## 7. Limitations and Outlook

On the basis of the theory of planned behavior and protection motivation, this paper analyzed the key factors of bad driving behavior of motor vehicle drivers. However, because of the shortcomings of the survey methods and the complex and diverse factors affecting individual drivers' bad driving behaviors, the paper still has some shortcomings, which need to be further improved and studied:

(1) This paper mainly adopted the questionnaire survey method for research, and there was a certain randomness when individual drivers filled in the questionnaire. Although unqualified questionnaires were eliminated according to scientific methods, it was difficult to check the survey accuracy of subjective intention of the questionnaire. In the future, a simulated driving method should be adopted to investigate bad driving behaviors, and the accuracy of survey data should be improved by summarizing the performance of drivers in actual operation and subsequent continuous observation.

(2) On the basis of the theory of planned behavior and the theory of protection motivation, this paper constructed the integration model of motor vehicle driver's bad driving behavior, and the data analysis results showed that the model and data fit well. Although this paper used behavioral and psychological factors to characterize external stimuli such as road traffic law enforcement environment and cost pressure, in fact, external stimuli should also include objective factors such as weather conditions, which need to be enriched in subsequent studies.

## 8. Conclusion

This paper summarized and classified drivers' bad driving behavior, and established the initial integrated structural equation model on the basis of the integration of TPB and PMT theories. This paper also selected the bad driving behavior influence variables through the questionnaire and used statistical methods and the structural equation model to analyze the statistical properties of the bad driving behavior, gaining the bad driving behavior of the key factors for novice drivers. This paper explored the decision-making process and influencing degree of factors of bad driving behavior from different perspectives, and provided a theoretical basis for early education, management intervention, and differential formulation of related rewards and punishments in the later period. There were two important aspects of this study, as follows:

(1) Integrating structural equation model of bad driving behavior based on behavior theory.

An integrated model of bad driving behavior including planned behavior theory and protective motivation theory was established. The integrated model was estimated on the basis of the partial least squares parameter estimation method. The behavior attitude, subjective norms, perceived behavior control, threat susceptibility, threat severity, internal and external returns, response cost, response efficiency, and self-efficacy are the key factors affecting drivers' bad driving behavior, and the causal relationship between the key factors was proven.

(2) Formulating targeted improvement strategies.

The integrated model was verified and analyzed by using the data of novice drivers. It found that the incentive and self-efficacy had the greatest impact on bad driving behavior intention. If the traffic management department can find and punish the bad driving behavior in time, the reward of the novice driver's compliance with traffic rules is greater than that of the bad driving behavior. In this way, the novice driver group will evolve into the trend of obeying traffic rules and reach a stable state of evolution. This not only reflects the influence of bad driving behavior on novice drivers, but also reflects the lack of relevant education, laws, and regulations for novice drivers.

**Author Contributions:** Conceptualization, L.Y. and X.Z.; methodology, X.Z; software, Y.L.; validation, L.Y. and X.Z.; investigation, X.Z. and Y.L.; resources, Y.L.; data curation, Y.L.; writing—original draft preparation, X.Z.; supervision, L.Y. and X.Z.; project administration, Y.L.

**Funding:** This research was funded by the Open Project of Key Laboratory of Ministry of Public Security for Road Traffic Safety, grant number 2019ZDSYSKFKT01.

**Acknowledgments:** The research for this paper has been supported by the Open Project of Key Laboratory of the Ministry of Public Security for Road Traffic Safety (no. 2019ZDSYSKFKT01). The authors sincerely thank all the teachers and classmates who gave many valuable suggestions on this paper.

**Conflicts of Interest:** The authors declare no conflict of interest.

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
