# Peer review of "Research on Risky Driving Behavior of Novice Drivers"

_sustainability, doi:10.3390/su11205556_

Round 1

Reviewer 1 Report

This article analyzes the risky driving behavior of novice drivers in China via the use of survey data and structural equation modeling.  It has the following weaknesses:

It is not clear what the specific contributions of this article are.  The majority of the findings and improvement strategies suggested are already known from the existing literature. This article is very poorly written; It has hundreds of grammatical and proof-reading errors. The authors must present any acronym in details before using it first (Abstract section: TPB & PMT). The authors used both British and American English terms.  For instance, behavior and behaviour. (Lines: 148-150 & 171-173) The sentences read very weirdly given that a specific word is used too many times.  The sentence in line 126. (Line 308) "clonal Bach." What do you mean by "law construction?" The statistics presented in the Introduction section must be supported by references. The citations of the references are not according to the journal guidelines. The authors used so many acronyms without presenting the meaning.  For instance, AAR, TRA, RP/SP, AMOS, etc. (Lines: 259-263) The descriptions presented are completely wrong.  The number of items described in the text does not conform to the item presented in Table 2. (Table 2)  Why are the item 13 and 14 same? What do you mean by berthing illegally in the context of road driving? (Table 6) Why do both <0.6 and >0.6 yields "OK" result?

Overall, this article is full of mistakes and wrong information without a proper description of the contributions.

Author Response

Point 1: It is not clear what the specific contributions of this article are.  The majority of the findings and improvement strategies suggested are already known from the existing literature.

Response 1: The specific contributions of this article are concluded  as followed : This paper combines the planned behavior theory (TPB) and the protection motivation theory (PMT), summarizes and classifies the bad driving behavior of novice drivers, and then studies the relationship between the bad driving behavior of novice drivers and their subjective factors, social cognitive factors and traffic education factors through case analysis and model construction. Different from previous studies, the formation mechanism of the behavior is discussed from the psychological point of view. It can help explain the risky driving behavior of novice drivers, as well as provide the theoretical basis for correcting bad driving behavior with strategies like early education and intervention, and bonus-punishment differentiation specifically for individuals.It can also improve road safety awareness of novice drivers and the awareness of compliance with traffic rules.

Point 2: This article is very poorly written; It has hundreds of grammatical and proof-reading errors. The authors must present any acronym in details before using it first (Abstract section: TPB & PMT). The authors used both British and American English terms.  For instance, behavior and behaviour. (Lines: 148-150 & 171-173) The sentences read very weirdly given that a specific word is used too many times. 

Response 2: TPB is abbreviated from the Theory of Planned Behavior. PMT is abbreviated from the Theory of Planned Behavior. The above questions have been modified and clearly highlighted in the original text.

Point 3: The sentence in line 126. (Line 308) "clonal Bach." What do you mean by "law construction?"

Response 3: Law construction means relevant governments and administrative departments make relevant laws and regulations to restrict or control drivers' driving behaviors, such as some criminal responsibilities.

Point 4:  The statistics presented in the Introduction section must be supported by references. The citations of the references are not according to the journal guidelines. 

Response 4: The statistics are from the statistical yearbook of the Traffic Management Bureau of the Ministry of Public Security of China. And the paper has modified the citations of the references.

Point 5: The authors used so many acronyms without presenting the meaning.  For instance, AAR, TRA, RP/SP, AMOS, etc.

Response 5: AAR is abbreviated from the Anticipated affective reactions. TRA is abbreviated from the Theory of Reasoned Action. RP is abbreviated from the Revealed Preference survey. SP is abbreviated from the Stated preference survey. AMOS is abbreviated from the Advanced MOrtar System. And some of them has been modified in the manuscript.

Point 6: (Lines: 259-263) The descriptions presented are completely wrong.  The number of items described in the text does not conform to the item presented in Table 2. (Table 2)  Why are the item 13 and 14 same? What do you mean by berthing illegally in the context of road driving? (Table 6) Why do both <0.6 and >0.6 yields "OK" result? 

Response 6: The above questions have been modified and clearly highlighted in the original text. It’s my careless on revision that caused these  questions and I would like to express my apologies for these questions.  >0.6 yields "OK" result. Berthing illegally means drivers  stop at the roadside at will not following regulations or sign indication.

Reviewer 2 Report

Much has been written about these topics.  Perhaps the authors might re-formulate their argument and focus on adding a new perspective to this area or further developing this area?

The biggest gap is the lack of references to literature. In many places the authors refer to factors that are perfectly known but without any references.

I suggest to work on literature and improve it significantly.

I also suggest you insert the limitation section

The conclusions should be harmonized and focus on the most important factors. Most of this can be moved to the previous section,

Author Response

Point 1: The biggest gap is the lack of references to literature. In many places the authors refer to factors that are perfectly known but without any references.I suggest to work on literature and improve it significantly.It also suggest you insert the limitation section.

Response 1: The above questions have been modified and clearly highlighted in the original text.And the limitation section has been added in the conclusion section.

Point 2: The conclusions should be harmonized and focus on the most important factors. Most of this can be moved to the previous section

Response 2: Revise the conclusion, analyze the most important factors of this paper, as well as the limitations of the model and the future direction.The above question has been modified and clearly highlighted in the original text.

Reviewer 3 Report

Dear Authors,
in my opinion, the article deals with interesting issues that are of a theoretical and methodological nature, as well as a large practical potential. For such studies, the balance between science and the expert approach is always the most important. I rate the reviewed article positively, both in the theoretical, methodological and empirical layer. However, there are some drawbacks in the article that should be corrected before it is published despite my positive review. First of all, the literature review is too modest and one should rely on a wider group of studies that have been carried out before.

Only 21 publications / studies were cited. It definitely did. It is necessary to deepen the literature studies and refer to other previously published studies.

It is also worth reviewing the text again for letter errors and minor language errors.

Author Response

Point 1: First of all, the literature review is too modest and one should rely on a wider group of studies that have been carried out before.Only 21 publications / studies were cited. It definitely did. It is necessary to deepen the literature studies and refer to other previously published studies.

Response 1: The manuscript has added some publications / studies.The above question has been modified and clearly highlighted in the original text.

Point 2: It is also worth reviewing the text again for letter errors and minor language errors.

Response 2: The above question has been modified and clearly highlighted in the original text.

Reviewer 4 Report

This study investigates the effects of different factors on bad driving behavior of novice drivers. I think this is an interesting idea, and could be in a modified form, a useful model for exploring novice drivers' behavior. Please see my comments as follows:

The authors should proofread the manuscript carefully.  Few errors are listed here: Figure 1: “Investigate andanalyzedata” Line 81: “to predict drivers'intentions for road” Line 101: “by personal behavioral beliefs,regulate beliefs and” Line 118: “novice drivers'bad driving" Table 6: “Attitudestowards behaviours” Line 309: “Finally, it is found that the internal consistency coefficient of the scale is shown in Table 6 below.”

2. Line 13: the sentence should be changed to “based on the theory of planned behavior (TPB) and the protection motivation theory (PMT)” since you are using these abbreviations in next sentence.

3. Line 143: Are the classifications in this section proposed in this study or should be referenced to previous studies?

4. Line 223: The term “RP/SP” should be defined

5. The process of random participant selection should be discussed. For example, how many streets were considered to distribute the questionnaires? Are the streets from different functional classes?

Author Response

Point 1: The authors should proofread the manuscript carefully.  Few errors are listed here: Figure 1: “Investigate andanalyzedata” Line 81: “to predict drivers'intentions for road” Line 101: “by personal behavioral beliefs,regulate beliefs and” Line 118: “novice drivers'bad driving" Table 6: “Attitudestowards behaviours” Line 309: “Finally, it is found that the internal consistency coefficient of the scale is shown in Table 6 below.”

Response 1: The above question has been modified and clearly highlighted in the original text.

Point 2: Line 13: the sentence should be changed to “based on the theory of planned behavior (TPB) and the protection motivation theory (PMT)” since you are using these abbreviations in next sentence.

Response 2: The  question has been modified and clearly highlighted in the original text.

Point 3: Line 143: Are the classifications in this section proposed in this study or should be referenced to previous studies?

Response 3: The classifications in this section are proposed in this study  according to different behavioral psychological states of drivers' risky driving behaviors.

Point 4:  Line 223: The term “RP/SP” should be defined

Response 4: RP is abbreviated from the Revealed Preference survey. SP is abbreviated from the Stated preference survey.And the  question has been modified and clearly highlighted in the original text.

Point 5: The process of random participant selection should be discussed. For example, how many streets were considered to distribute the questionnaires? Are the streets from different functional classes?

Response 5: This survey adopted simple random sampling to collect specific personal attributes, cognitive status information of bad driving behaviors and characterization information of bad driving behaviors from motor vehicle drivers through online answering papers and roadside inquiries.

Round 2

Reviewer 1 Report

The authors have responded to some of the reviewer's comments but not all.  Even after careful reading several times, the reviewer couldn't follow in some of the responses what the authors were trying to convey.  In the revised version of the paper, the grammatical and proof-reading issues became more severe.  For instance,

Line: 20 "Additionally, we analysebased on the modelling integration of"

Line: 41 "newAccording to statistics, if the number of traffic accidents is divided by driver's driving age,"

Line: 49 "This paper, we study combines the planned behavior theory (TPB) and the protection"

...

There are hundreds of the above issues present in the paper.  Furthermore, it is very hard to follow what revisions the authors have performed.

Author Response

Dear reviewer:
Thanks for your kind advice . According with your advice, we amended the relevant part in manuscript. And your questions were answered below.

Point 1: In the revised version of the paper, the grammatical and proof-reading issues became more severe. There are hundreds of the above issues present in the paper. 

Response 1: First of all, I would like to express my apologies for the excessive grammatical and spelling mistakes in the article. After checking, I found that the manuscript I saved is different from what you said in the comments. This may be a problem with the revision function in word or an error occurred when uploading the system. I have told the editor about this problem. This time I cancell the revision function in word and present all the modified content to you in yellow background. I express my apologies again.

Point 2: Even after careful reading several times, the reviewer couldn't follow in some of the responses what the authors were trying to convey. Furthermore, it is very hard to follow what revisions the authors have performed.

Response 2: According to your revision suggestions last time, I have sorted out the main contributions of the manuscript as follows:From the perspective of behavior theory, this paper analyzes the causes of drivers' risky driving behavior, and constructs an integrated model of motor vehicle drivers' bad driving behavior based on the theory of planned behavior and the theory of protection motivation. The objective is to quantify and analyze the formation mechanism of drivers' risky driving behaviors, effectively explain the characteristics of drivers' bad driving behaviors, and provide theoretical basis for the early education, management intervention and the formulation of related rewards and punishment policies and measures in the later stage of bad driving behaviors. The acronym in the paper were all marked with full names. All modifications were also presented in yellow background.And I integrated the literature review according to the dangerous driving behavior and TPB and PMT theory this time and added several references.

Thanks for your kind advice again. 

Reviewer 2 Report

The authors have creatively utilized secondary sources and made a good effort to revise the second submission taking into account the reviewer comments but the paper still need few improvements

The authors did not answer my question about re-formulate their argument and focus on adding a new perspective to this area or further developing this area There is still some work to be done to improve its value to readers of Sustainability with references Limitations should be separated as a independent section A short, introductory paragraph summarizing the intent and scope of the study would provide a useful context for the rest of the paper. In addition, a closing paragraph or two with summative insights, or overarching principles garnered from this research would make the manuscript more complete. As written, the final paragraphs seem an abrupt end to the work and the paper lacks a sense of closure.

Author Response

Dear reviewer:
Thanks for your kind advice . According with your advice, we amended the relevant part in manuscript. And your questions were answered below.

Point 1: The authors did not answer my question about re-formulate their argument and focus on adding a new perspective to this area or further developing this area.

Response 1: I summarized the characteristics of the existing research on motor vehicle driver's bad driving behavior at home and abroad and the application of relevant behavioral theories, and determined the research direction of the manuscript. The revision of this part is reflected in the literature review.

Point 2: There is still some work to be done to improve its value to readers of Sustainability with references.

Response 2: I integrated the literature review according to the dangerous driving behavior and TPB and PMT theory this time and added several references. Literature review section has illustrated the influence factors on novice driving behavior, the influence factors on dangerous driving behavior and the relevant application of TPB and PMT theory on driving behavior by the relevant scholars.

Point 3: Limitations should be separated as a independent section. A short, introductory paragraph summarizing the intent and scope of the study would provide a useful context for the rest of the paper. In addition, a closing paragraph or two with summative insights, or overarching principles garnered from this research would make the manuscript more complete. As written, the final paragraphs seem an abrupt end to the work and the paper lacks a sense of closure.

Response 3: I set the limitation and outlook section as section 7 and rewrite this section. The introductory paragraph is also adjusted and the conclusion is revised from three aspects.

All modifications above are marked with yellow background.

Thanks for your kind advice again.

Round 3

Reviewer 1 Report

Please do another round of proof-reading.